# Phytochemical Characterization of *Cannabis sativa* L. Chemotype V Reveals Three New Dihydrophenanthrenoids That Favorably Reprogram Lipid Mediator Biosynthesis in Macrophages

**DOI:** 10.3390/plants11162130

**Published:** 2022-08-16

**Authors:** Stefano Salamone, Lorenz Waltl, Anna Pompignan, Gianpaolo Grassi, Giuseppina Chianese, Andreas Koeberle, Federica Pollastro

**Affiliations:** 1Department of Pharmaceutical Sciences, University of Piemonte Orientale, 28100 Novara, Italy; 2PlantaChem Srls, 28100 Novara, Italy; 3Michael Popp Institute and Center for Molecular Biosciences Innsbruck (CMBI), University of Innsbruck, 6020 Innsbruck, Austria; 4Canvasalus Srl, 35043 Monselice, Italy; 5Department of Pharmacy, University of Naples Federico II, 80131 Naples, Italy

**Keywords:** fiber hemp, chemotype V, dihydrophenanthrene, lipid mediator, Ermo

## Abstract

The growing general interest surrounding *Cannabis sativa* L. has led to a renewal in breeding and resulted in an impressive variability of chemotypical characteristics that required the division of cannabis into different recognized chemotypes. The chemotype V has been overlooked in terms of phytochemical composition due to the almost total absence of cannabinoids, on which biomedical attention is focused. Systematic approaches addressing diverse chemotypes are, however, needed to discriminate and define phytochemical aspects beyond cannabinoids. Such thoroughly characterized chemotypes guarantee blinding in controlled studies by mimicking the sensory properties of hemp and may help to unravel the “entourage effect”. Capitalizing on the ability of cannabis to synthesize a large number of non-cannabinoid phenolic compounds, we here investigated, for the first time, the composition of the Ermo chemotype V and identified new compounds: two dihydrophenanthrenes and the methoxy-dihydrodenbinobin. All three compounds suppress pro-inflammatory leukotriene biosynthesis in activated macrophage subtypes by targeting 5-lipoxygenase, but substantially differ in their capacity to elevate the levels of specialized pro-resolving lipid mediators and their precursors in M2 macrophages. We conclude that the discovered compounds likely contribute to the anti-inflammatory properties of *Cannabis sativa* L. chemotype V and might promote inflammation resolution by promoting a lipid mediator class switch.

## 1. Introduction

*Cannabis sativa* L. was one of the first plants ever domesticated by humans, who spread its cultivation worldwide over the past 10,000 years. Due to its considerable and heterogeneous use, the plant can be considered as a multi-purpose crop, an object of constant breeding and selection [1]. The harvesting of cannabis has benefited from continuous developments and has been incredibly revived at the industrial level in several European countries such as Italy, Spain, Germany, the Netherlands and France [2]; this is also due to the progressive liberalization of non-psychoactive cultivars and their ongoing phytochemical analysis. More than 500 secondary metabolites have been identified, comprising cannabinoids, terpenoids and phenolic compounds [3].

The increasing public interest in cannabis together with the impressive variability of chemotypical characteristics inspired the division of cannabis into five recognized groups (or chemotypes) within the genus, depending on different concentrations of major cannabinoids (Δ^9^-THC **1**; CBD **2**; and CBG **3**). The five recognized chemotypes of cannabis include: (I) the drug-type plants (narcotic) with a high content of the psychotropic Δ^9^-tetrahydrocannabinol (Δ^9^-THC **1**); (II) medicinal cannabis with a 1:1 content of Δ^9^-THC **1**: CBD **2**; (III) industrial fiber hemp that has CBD **2** as a predominant constituent and a minimum content of Δ^9^-THC **1** (0.2% *w*/*w*); (IV) fiber-type plants that contain cannabigerol (CBG **3**) as the main cannabinoid; (V) fiber-type plants largely devoid of cannabinoids (Table 1) [4]. The latter cannabinoids are certainly spectacular secondary metabolites that have long been in the spotlight of biomedical attention. However, it is also important to discriminate and define the qualitative and quantitative aspects of other chemotypes beyond cannabinoids. Such a systematic approach allows further insights into bioactive constituents, such as cannflavins, which might synergistically contribute to the anti-inflammatory activity of cannabis [1,5]. Moreover, cannabinoid-free strains are necessary for controlled studies to mimic the sensory properties of hemp, guarantee blinding and explain the typical synergy of many botanical extracts besides the isolated cannabinoids [5]. 

Of all the chemotypes, type V (cannabinoid-free) has been overlooked in terms of phytochemical composition because cannabinoids are absent. Many hypotheses have arisen regarding this inability to produce cannabinoids. For example, the absence of this biosynthesis could be due to the disabling of terpeno-phenolic condensation or a total absence and dysfunctionality of glandular trichomes. The presence of a cannabinoid knock-out factor that leaves unaffected the biosynthesis of other compound classes is decisive in inactivating the pathway toward the phenolic cannabinoid precursors [6]. Anyway, cannabis is a prolific producer of metabolites and besides cannabinoids, the plant biosynthesizes other molecules, such as terpenoids responsible for the typical scent, oxylipins, amines and amides, phytosterols and a plethora of non-cannabinoid phenolic compounds [7]. Among non-cannabinoid phenols, denbinobin **4** deserves particular mention for its biological activity and for its recurrence, which is extremely rare in the plant kingdom. Denbinobin **4**, a typical ingredient of several orchidaceous plants, has been identified in the IV chemotype of fiber-hemp [8]. Its anti-tumor potential has cogently been conveyed by the title of a commentary in the *British Journal of Pharmacology*: Escaping immune surveillance in cancer: is denbinobin the panacea? [9]. Other unique compounds of cannabis are canniprene **6** and cannflavins **7a** and **8**, which open a new field for chemical and biological exploration [1,10]. In particular, canniprene **6** potently inhibits the production of inflammatory eicosanoids via the 5-lipoxygenase (5-LOX) pathway [11]. It outperforms the structural analogue resveratrol in suppressing pro-inflammatory readouts in diverse cell-free test systems and shows a remarkable potential for application in skin care [12]. The prenylated lipophilic flavonoids, cannflavins **7a** and **8**, are potent inhibitors of inducible inflammatory enzymes such as microsomal prostaglandin E2 synthase (mPGES)-1 and act as “intelligent” suppressors of the inflammatory response [12].

Given the huge number of bioactive compounds beyond cannabinoids, we were interested in the non-cannabinoid phenolic profile present in the V chemotype, the anti-inflammatory activity of individual components, and the impact of breeding on the chemical and functional fingerprint. For this reason, we here investigated, for the first time, the composition of the Ermo fiber hemp (a variety of hemp with almost no cannabinoids) provided by Canvasalus and reported on the isolation, phytochemical characterization, and biological evaluation of phenolic compounds, of which three derivatives are new. Our phytopharmacological studies employed a targeted metabololipidomics approach and focused on molecular targets within anti-inflammatory, pro-resolving and immunomodulatory lipid mediator biosynthesis pathways.

## 2. Results and Discussion

The analysis of the phytochemical profile of the Ermo chemotype V led to the isolation of trace amounts (0.14%) of cannabidiolic acid **2a** and cannabigerolic acid **3a**. Chromatographic purification yielded non-cannabinoid phenolic compounds, i.e., the dihydrostilbenoid canniprene **6**, the C-prenylated flavonoids cannflavin A **7a**, cannflavin B **8** and 5′-demethoxy-cannflavin A **7**, which was first identified in *Morus alba* var. *tatarica* in 2015 [13] and only recently in *Cannabis sativa* [14]. The Ermo fiber-hemp provided three natural compounds **5, 9** and **10**, which have never been isolated before and belong to the biosynthetic group of dihydrophenanthrenoids (Figure 1).

Compound **9,** molecular formula determined by HR-ESIMS to be C_17_H_19_O_5_ with *m/z* 303.12, is a dihydrophenanthrene isolated as a brown powder. The ^1^H NMR spectroscopic data (Table 2 and Appendix A) displayed signals due to three aromatic protons (δ_H_ 6.62, s (H-1), 6.44, d (*J* = 2.70 Hz, H-6), 6.50, d (*J* = 2.70 Hz, H-8)), two methylene moieties (δ_H_ 2.64, bs (H-9 and H-10)) and three methoxy groups (δ_H_ 3.84, s (OMe-2), 3.63, s (OMe-3), 3.81, s (OMe-7)), suggesting the structure of 9,10-dihydro-2,3,7-trimethoxy-4,5-phenanthrenediol (Figure 2 and Table 2). This structure was confirmed by ^13^C NMR spectroscopy (Table 3), which revealed the presence of four aromatic quaternary carbons (δ_C_ 117.3 C-4a, 113.3 C-4b, 127.5 C-8a and 142.7 C-10a) and five oxygenated aromatic quaternary carbons (δ_C_ 151.3 C-2, 137.0 C-3, 150.3 C-4, 155.7 C-5, 160.4 C-7), three aromatic methine carbons (δ_C_ 99.1 C-1, 102.0 C-6, 106.4 C-8), two methylene groups (δ_C_ 31.2 C-9, 21.8 C-10) and three methoxyls (δ_C_ 55.3 OMe-2, 61.2 OMe-3, 54.5 OMe-7). Methoxy group locations at C-2, C-3 and C-7 were confirmed by the HMBC and NOESY correlations. Specifically, NOESY correlations observed between the methoxy group at δ_H_ 3.84 and the aromatic proton at H-1 (δ_H_ 6.62) together with the NOESY observed between the two methoxy groups at δ_H_ 3.84 and δ_H_ 3.63, placed the two methoxy groups, respectively, at positions C-2 and C-3. This was further confirmed by the HMBC observed between H-1 and C-2. Moreover, HMBC correlations (H-6/C-4b, C-5, C-7, C-8; H-8/C-4b, C-7, C-9; OMe-7/C-7) and NOESY correlations (H-6/OMe-7 and H-8/OMe-7) supported the assignment of the remaining methoxy group at C-7 (Figure 2 and Appendix A).

Compound **10**, molecular formula determined by HR-ESIMS to be C_17_H_19_O_5_ with *m/z* 303.12, isolated as a brown powder, is closely related to compound 9 (Figure 2), with the ^1^H and ^13^C NMR spectroscopic data (Table 2 and Table 3 and Appendix A) indicating the same dihydrophenanthrene structure that displays different disposition of aromatic oxygenated carbon atoms. In compound 10, aromatic protons form one singlet at δ_H_ 6.63 (H-1) and two doublets at δ_H_ 6.87 (*J*_H_ = 8.2 Hz, H-7) and 6.82 (*J*_H_ = 8.2 Hz, H-8), which were ascribed to a pair of ortho-coupled protons, while the three methoxy groups are singlets at δ_H_ 3.87 (OMe-2), 3.79 (OMe-4) and 3.86 (OMe-6). The ^13^C spectrum (Appendix A) shows the presence again of five oxygenated aromatic quaternary carbons (δ_C_ 152.1 C-2, 147.4 C-3, 136.6 C-4, 141.5 C-5, 147.6 C-6). The HMBC correlations (Appendix A) from H-7 to C-5, C-6, C-8, C-8a, from H-8 to C-4b, C-6, C-7, C-8a, C-9 and from H-6 to the carbon on OMe-6 together with the NOESY correlation OMe-6/H-7 allowed the identification of the ring C substructure. The remaining two methoxy groups have been assigned based on the HMBC correlations (Appendix A) OMe-2/C-2 and OMe-4/C-4 and the NOESY correlation between only OMe-2 and H-1. Based on the above-described evidence, the structure of **10** was defined as 9,10-dihydro-2,4,6-trimethoxy-3,5-phenanthrenediol (Figure 2).

Compound **5** was obtained as a red powder. Its molecular formula was determined by HR-ESIMS to be C_17_H_17_O_5_ with *m/z* 301.10 (Appendix A). In the ^1^H NMR spectrum (Table 2, Appendix A), there are only two aromatic protons (δ_H_ 6.54, s (H-6), 6.53, s (H-8)) and one olefinic proton (δ_H_ 5.95, s (H-2)). Moreover, two methylene moieties (δ_H_ 2.67, t (H-9), 2.51, t (H-10)) and three methoxyls (δ_H_ 3.87, s (OMe-3), 3.76, s (OMe-5), 3.85, s (OMe-7)) are still present. The ^13^C spectrum (Appendix A) showed two carbonylic carbons (δ_C_ 185.2 C-1, 179.2 C-4) and confirmed the presence of three sp^2^ oxygenated carbon atoms (δ_C_ 160.1 C-3, 158.0 C-5, 162.0 C-7), while there are still four olefinic disubstituted carbon atoms, two ascribable to the aromatic ring (δ_C_ 112.0 C-4b, 138.8 C-8a) and the others to the quinonoid moiety (δ_C_ 138.0 C-4a, 140.1 C-10a) and two methines (δ_C_ 28.3 C-9, 20.1 C-10).

This evidence suggests that compound **5** is a dihydrophenanthrenequinone. The key HMBC correlations (Appendix A) of H-10 to C-1, C-3, C-4, C-10a; H-6 to C-4b, C-5, C-7, C-8; H-8 to C-4b, C-6, C-7, C-9; H2-10 to C-1; OMe-3 to C-3; OMe-5 to C-5 and OMe-7 to C-7 supported this assignment. Furthermore, locations of methoxy groups at C-3, C-5 and C-7 were confirmed by the NOESY correlations H-2/OMe-3, H-6/OMe-5 and OMe-7, H-8/OMe-7 (Appendix A). On the basis of these results, compound **5** was identified as 9,10-dihydro-3,5,7-trimethoxy-1,4-phenanthrenequinone or a new 5-methoxy-dihydrodenbinobin derivative **5** (Figure 2).

Even though a lot of work has been conducted on the phytochemical characterization of cannabis, this is the very first time the non-cannabinoid compounds **5**, **9** and **10** have been isolated and characterized from this plant. The dihydrophenanthrenes **9** and **10** are particularly interesting for a possible structure–activity relationship study, since both compounds retain the same number of methoxy and hydroxyl groups but with different positions in the reciprocal scaffold. Moreover, just a few cannabis phenanthrenes have so far been described with properties that might be relevant for human health, such as the unique denbinobin **4** endowed with cell apoptosis-inducing properties that might rely on the impaired activation of the survival kinase Akt (protein kinase B) [15,16] or the interference with NF-kB signaling [17]. Given the biological relevance of denbinobin **4**, the newly isolated methoxy-reduced denbinobin analogue **5** described here, is expected to be of similar biological relevance.

Diverse phenanthrene secondary metabolites exert anti-inflammatory, anti-allergic, and/or anti-platelet aggregation activity, among others by interfering with lipid mediator biosynthesis [18,19]. To assess the effect of the three new dihydrophenanthrenoids **5, 9** and **10** on the lipid mediator network, we challenged human M1 and M2 macrophages with *Staphylococcus aureus*-conditioned medium and analyzed major pro-inflammatory, anti-inflammatory and pro-resolving lipid mediators by targeted metabololipidomics (Figure 3a,b). Compounds **9** and **10** exhibited anti-inflammatory effects in human M1 macrophages by inhibiting 5-LOX product formation at low micromolar concentrations. (Figure 3c). The quinone **5** was most potent (IC_50_ = 1.0 µM) and the OH-methylation pattern modulated the 5-LOX product-lowering activity (**9**: IC_50_ = 3.8 µM, **10**: IC_50_ = 8.5 µM). Direct inhibition of human recombinant 5-LOX was confirmed for compounds **9** (IC_50_ = 6.9 µM) and **10** (IC_50_ = 4.6 µM) in a cell-free assay. Compound **5** was less active on isolated 5-LOX (IC_50_ = 20.1 µM) (Figure 3d,e), despite being the most potent inhibitor of 5-LOX product formation in intact cells (IC_50_ = 1 µM) (Figure 3c,e), which hints either towards cellular accumulation or alternative targets within the 5-LOX pathway, such as the 5-LOX-activating protein FLAP [20]. In addition, quinone **5** suppressed the formation of overall pro-inflammatory cyclooxygenase (COX)-derived prostanoids (IC50 = 2.0 µM), whereas the diols **9** and **10** substantially elevated COX product formation in M1 macrophages (Figure 3c), probably due to the redirection of arachidonic acid from the 5-LOX to the COX pathway [20].

In human M2 macrophages, which are associated with wound healing and tissue repair [21,22] compounds **9** and **10** stimulated the production of specialized pro-resolving mediators (SPMs) (Figure 3b,f), which actively promote the termination of inflammation and return to homeostasis by inhibiting neutrophil trafficking, stimulating efferocytosis, enhancing bacterial clearance, protecting from oxidative stress, and promoting tissue regeneration [23,24,25]. Interestingly, compound **9** raised the levels of multiple SPM classes, i.e., D-series Rv (RvDs), protectins (PDs) and maresin 2 (MaR2), whereas the isomeric compound **10** selectively increased the availability of MaR2, and the quinone **5** elevated MaR2 and PDs but not RvDs levels (Figure 3b,f). It is tempting to speculate that the 3- and/or 7-methoxy-groups, which are shared by **9** and **5,** but not present in **10,** in these isolated compounds, are required for an effective increase in PDs levels. Moreover, PDs and MaR2 were strongest upregulated by compound **5** (Figure 3b,f); although, the release of polyunsaturated fatty acids (PUFAs) was decreased (Figure 3g), which indicates that the quinone substructure is not only advantageous for 5-LOX inhibition and required for suppressing COX product formation but is also favorable for the lipid mediator class switch from pro-inflammatory LTs and prostanoids to pro-resolving SPMs.

Mechanistically, the overall increase in SPM levels induced by **9** (Figure 3b,f) seems to arise from an enhanced formation of precursors, including 17-HDHA, 15-HEPE and 15-HETE (Figure 3b) by 15-LOX-dependent oxygenation of docosahexaenoic acid (DHA) [26,27]. Compound **5**, on the other hand, slightly but significantly lowered 12/15-LOX product levels (Figure 3h), which rather excludes that 12- and 15-LOX drive the upregulation of PDs and MaR2 (Figure 3f). Of interest, soluble epoxide hydrolase (sEH) not only hydrolyzes epoxyeicosatrienoic acids (EETs) to their corresponding diols (DHETs) but also converts 13(S), 14(S)-epoxy-DHA to MaR2 [28]. Since the ratio of DHETs/EETs, which serves as an indicator of sEH activity [29], is significantly elevated by compound **5** in human M2 macrophages (Figure 3i), it is tempting to speculate that the higher sEH activity adds to the associated increase in MaR2 and potentially PDs levels (Figure 3f). Compound **10** evokes comparable but less pronounced effects (Figure 3b,f); although, sEH activity is only increased by trend (Figure 3i). Together, the newly identified dihydrophenanthrenoids promote a lipid mediator class switch from inflammation to resolution, with small structural changes deciding about the lipid mediator subgroups affected.

## 3. Material and Method

### 3.1. General Experimental Procedures

^1^H (600 MHz, 400 MHz) and ^13^C (150 MHz, 100 MHz) NMR spectra were measured on Bruker Avance 700 spectrometer and Bruker 400 spectrometers (Bruker^®^, Billerica, MA, USA). Chemical shifts were referenced to the residual solvent signal (C_3_D_6_O: δ_H_ = 2.05, δ_C_ = 206.7, 29.9). Homonuclear ^1^H connectivities were determined by the COSY experiment. One-bond heteronuclear ^1^H−^13^C connectivities were determined with the HSQC experiment. Two- and three-bond ^1^H−^13^C connectivities were determined by gradient 2D HMBC experiments optimized for a 2,3 *J* = 9 Hz. Low- and high-resolution ESIMS data were obtained on an LTQ OrbitrapXL (Thermo Scientific, Waltham, MA, USA) mass spectrometer. Silica gel 60 (70−230 mesh), RP C-18 silica gel and Celite^®^ 545 particle size 0.02–0.1 mm, pH 10 (100 g/L, H_2_O, 20 °C), used for low-pressure chromatography and vacuum chromatography was purchased from Macherey-Nagel (Düren, Germany). Purifications were monitored by TLC on Merck 60 F254 (0.25 mm) plates, visualized by staining with 5% H_2_SO_4_ in EtOH and heating. Chemical reagents and solvents were from Aldrich (Darmstadt, Germany) and were used without any further purification unless stated otherwise. Flash chromatography Isolera One with DAD (Uppsala, Sweden), HPLC JASCO Hichrom, 250 × 25 mm, silica UV−vis detector-2075 plus (Oklahoma, Japan).

### 3.2. Plant Material

*Cannabis sativa* L. Ermo variety was purchased from Canvasalus Srl (Monselice, Italy). A voucher specimen (CsErmo-2022) of the vegetal material is stored in Novara laboratories. The variety is protected at Community Plant Variety Office (CPVO) with application number 20100208.

### 3.3. Extraction and Isolation

Nonwoody aerial parts, inflorescences and leaves (954 g) were extracted with acetone (2 × 10 L) in a vertical percolator at room temperature, affording 46.18 g (4.84%) of a dark green syrup. This was later dissolved at 45 °C in 400 mL of MeOH (ratio extract/MeOH 1:10) and left at 8 °C to condense fatty acids and waxes. After 12 h the solution was vacuum-filtered with cold MeOH in a sintered funnel protected by a bed of stratified Celite obtaining 20 g of residual fraction after evaporation with rotary evaporator. This latter part was subsequently purified by solid-phase extraction on RP C-18 silica gel to remove pigments, unsaturated fatty acids and poly-isoprenoids. For this purpose, the fraction was dissolved in the minimal MeOH amount at 45 °C and charged on 200 g of RP-C18 (ratio extract/RP-C18 1:10) packed with MeOH in a sintered funnel (9 × 15 cm) with a side arm for vacuum. Elution with MeOH (100 mL) gave 10 g of purified fraction. The latter fraction was fractionated by low-pressure chromatography (LPC) on silica gel (200 g, petroleum ether−EtOAc gradient from 90:10 to 20:80) to afford three fractions (I, II, and III), which were further purified. Fraction I was fractionated by flash chromatography with Isolera One on RP C-18 silica gel (60 g, solvent A: MeOH 0.03% formic acid, solvent B: H_2_O 0.03% formic acid gradient from 50:50 to 95:5) to afford 1.32 g of a mixture of cannabidiolic acid and cannabigerolic acid (**2a** and **3a**, 0.14%) as brown syrup, and 82 mg of canniprene (**6**, 0.008%) as a brownish powder. The separation of fraction II by with Isolera One on RP C-18 silica gel (12 g, solvent A: MeOH 0.03% formic acid, solvent B: H_2_O 0.03% formic acid gradient, from 50:50 to 95:5) followed by purification with HPLC on silica gel (petroleum ether-EtOAc gradient, from 60:40 to 30:70) afforded 37 mg of 9,10-dihydro-2,3,7-trimethoxy-4,5-phenanthrenediol (**9**, 0.005%) as a brownish powder, 36 mg of 9,10-dihydro-3,5,7-trimethoxy-1,4-phenanthrenequinone (**5**, 0.004%) as a red powder and 35 mg of 9,10-dihydro-2,4,6-trimethoxy-3,5-phenanthrenediol (**10**, 0.004%) as a brownish powder. Fraction III was fractionated by Isolera One on RP C-18 silica gel (12 g, solvent A: MeOH 0.03% formic acid, solvent B: H_2_O 0.03% formic acid gradient, from 50:50 to 95:5) and further separated by HPLC on silica gel (petroleum ether-EtOAc gradient, from 60:40 to 30:70) to afford 46 mg of cannflavin A (**7a**, 0.005%) as a yellow powder, 17 mg of cannflavin B (**8**, 0.002%) as yellow powder, 4 mg of 5′-demethoxy-cannflavin A (**7**, 0.0004%) as a yellow powder.

### 3.4. Spectroscopic Data

Compound **9**. Brown amorphous powder. ^1^H NMR (C_3_D_6_O, 400 MHz) and ^13^C NMR (C_3_D_6_O, 100 MHz): Table 1 and Table 2; HR-ESIMS found *m/z* 303.1230 [M + H]^+^; C_17_H_19_O_5_ requires *m/z* 303.1227.

Compound **5**. Red amorphous powder. C_3_D_6_O, 400 MHz) and ^13^C NMR (C_3_D_6_O, 100 MHz): Table 1 and Table 2; HR-ESIMS found *m/z* 301.1066 [M + H]^+^; C_17_H_17_O_5_ requires *m/z* 301.1076.

Compound **10**. Brown amorphous powder. ^1^H NMR (C_3_D_6_O, 400 MHz) and ^13^C NMR (C_3_D_6_O, 100 MHz): Table 1 and Table 2; HR-ESIMS found *m/z* 303.1230 [M + H]^+^; C_17_H_19_O_5_ requires *m/z* 303.1227.

### 3.5. Isolation of Peripheral Blood Mononuclear Cells (PBMC) from Human Blood

Leukocyte reduction system chamber (LRSC) filters were provided by the Central Institute for Blood Transfusion and Immunological Department of Tirol Kliniken GmbH (Austria) with the informed consent of the volunteers. Only healthy blood donors between 18 and 65 years of age without medication for chronic diseases, fever, or deficiency symptoms and after physical examinations by trained medical personnel were included in the study. Human peripheral blood mononuclear cells (PBMC) were freshly isolated from LRSC filters. After dilution of the cell concentrates in PBS pH 7.4 containing 12.5 mM citrate and 14 mM glucose (130 mL, 37 °C), immune cells were separated via isopycnic density gradient centrifugation (400× *g*, 20 min, RT) using Histopaque^®^-1077 (Sigma-Alrich, St. Louis, MO, USA). PBMC were obtained from the interphase after hypotonic lysis of erythrocytes using water (2–5 mL) and two consecutive washing steps with PBS pH 7.4 (50 mL) (270× *g*, 5 min, RT).

### 3.6. Monocyte Differentiation and Macrophage Polarization

To differentiate monocytes, which are the predominant cell type in PBMC, towards M0 macrophages, they were first stimulated with 20 ng/mL GM-CSF or M-CSF (HiSS Diagnostics GmbH, Freiburg, Germany) for 6 days in macrophage medium (RPMI 1640 supplemented with 10% FCS, 2 mmol/L L-glutamine, 100 U/mL penicillin and 100 μg/mL streptomycin). These monocyte-derived macrophages were polarized for another 48 h in macrophage medium with 100 ng/mL lipopolysaccharide and 20 ng/mL interferon-γ (Peprotech, Hamburg, Germany) to obtain M1 or with 20 ng/mL interleukin-4 (Peprotech, Hamburg, Germany) to obtain M2 phenotypes [30,31].

### 3.7. Sample Preparation and Metabololipidomic Profiling of Lipid Mediators

Lipid mediator biosynthesis in human M1/M2 macrophages (1.5 mL PBS pH 7.4 plus 1 mM CaCl_2_) was stimulated with *Staphylococcus aureus* (LS1)-conditioned medium (1.0%, 3 h) after 15 min preincubation with DMSO (0.1%) or test compounds and stopped by addition of ice-cold MeOH (2.5 mL) containing deuterium-labeled internal standards (200 pg d8-5S-hydroxyeicosatetraenoic acid (HETE), d4-leukotriene (LT)B_4_, d5-lipoxin (LX)A_4_, d5-resolvin (Rv)D_2_, d4-prostaglandin (PG)E_2_, and 2000 pg d8-arachidonic acid (Cayman Chemical, Ann Arbor, MI, USA)) [32]. Samples were incubated at −20 °C for at least 1 h to precipitate proteins and then centrifuged (750× *g*, 10 min, 4 °C). The supernatant was mixed with acidified water (7 mL, pH 3.5) and loaded onto solid phase cartridges (Sep-Pak^®^ Vac 6cc 500 mg/6 mL C-18 (Waters, Milford, MA, USA)), which were conditioned with MeOH (6 mL) and equilibrated with water (2 mL). Columns were washed with water (6 mL) and hexane (6 mL, 4 °C) before lipid mediators were eluted with methyl formiate (6 mL). The organic phase was evaporated to dryness using a TurboVap LV (Biotage, Uppsala, Sweden) and the remaining lipids were dissolved in MeOH/water (1:1), centrifuged twice (21,100× *g*, 4 °C, 5 min), and subjected to UPLC-MS/MS analysis. Chromatographic separation of lipid mediators was carried out at 55°C on an Acquity UPLC BEH C-18 column (130Å, 1.7 μm, 2.1 × 100 mm, Waters) using an ExionLC AD UHPLC system (Sciex, Framingham, MA, USA). The gradient of mobile phase A (water/MeOH, 90/10, 0.01% acetic acid) and mobile phase B (MeOH, 0.01% acetic acid) was ramped at a flow rate of 0.35 mL/min from 35.6% to 84.4% B within 12.5 min followed by 5 min of isocratic elution at 97.8% B. Lipid mediators and fatty acids were analyzed in the negative ion mode by scheduled multiple reaction monitoring (detection window: 120 s) using a QTRAP 6500^+^ Mass Spectrometer (Sciex), which was equipped with an electrospray ionization source. The curtain gas was set to 40 psi, the collision gas to medium, the ion spray voltage to −4000 V, the heated capillary temperature to 500 °C, and the sheath and auxiliary gas pressure to 40 psi. Transitions selected for quantitation and the corresponding declustering potential (DP), entrance potential (EP), collision energy (CE) and collision cell exit potential (CXP) are listed in Table 4. Absolute lipid quantities refer to an external standard calibration and were normalized to a subclass-specific deuterated internal standard as well as cell numbers. Mass spectra were acquired and processed using Analyst 1.7.1 (Sciex) and Analyst 1.6.3 (Sciex), respectively.

### 3.8. Determination of 5-Lipoxygenase Activity

Human recombinant 5-LOX (Cayman Chemical, 10 U) was pre-treated with vehicle (DMSO, 0.1%) or test compounds for 10 min in 1 mL PBS pH 7.4, 1 mM EDTA and 1 mM ATP on ice before CaCl_2_ (2 mM) and arachidonic acid (20 µM) were added [33]. After 10 min at 37 °C, the reaction was stopped by the addition of ice-cold MeOH (1 mL) and PGB_1_ (2 ng) as internal standard. Clean-Up C-18 Endcapped SPE cartridges (100 mg, 10 mL, UCT, Bristol, PA, USA) were conditioned with MeOH (1 mL, twice) and equilibrated with water (1 mL). After acidification (530 µL PBS plus 60 mM HCl) and centrifugation (750× *g*, 10 min, 4 °C), supernatants containing lipid mediators were loaded onto the columns, which were first washed with water (1 mL) and then MeOH/water (25:75, 1 mL). After elution of lipid mediators with MeOH (300 µL), water (120 µL) was added and samples were centrifuged (21,100× *g*, 10 min, 4 °C) and subjected to LC-PDA analysis. 5-LOX products, including all trans-isomers of LTB_4_ and 5-HETE, were separated at 40 °C and a flow rate of 0.45 mL/min on a Kinetex C-18 LC column (100Å, 1.3 μm, 2.1 × 50 mm, Phenomenex, Torrance, CA) using a Nexera X2 UHPLC system (Shimadzu, Kyoto, Japan). The step gradient of mobile phase A (water/MeOH, 50/50, 0.05% trifluoroacetic acid) and mobile phase B (MeOH, 0.05% trifluoroacetic acid) was kept at 14% B for 2 min, followed by 46% B for 2 min and 90% B for another 2 min. LTB_4_ isomers and 5-HETE were detected with a photodiode array detector (SPD-M20A, Shimadzu, Kioto, Japan) at 280 nm and 235 nm, respectively. Lipid quantitieswere calculated by internal calibration using PGB1 as reference standard.

### 3.9. Statistics

Data are expressed as mean ± s.e.m. and single data from n = 4–8 independent experiments. Significant outliers were removed (Grubbs’test, *p* < 0.05) and data were log-transformed for statistical analysis. For multiple comparisons, repeated-measures one-way ANOVA followed by Dunnett´s post hoc test was performed. *p* values < 0.05 were considered statistically significant. Data were analyzed using Microsoft Excel (Office 365, Version: 2204, Albuquerque, New Mexico), and statistics were performed using GraphPad Prism (GraphPad Software version 9.3.1, Dotmatics, Boston, UK). IC_50_ values were calculated by non-linear regression (GraphPad Prism 9.3.1).

## 4. Conclusions

The Ermo strain of *Cannabis sativa* belongs to the chemotype V and is almost completely devoid of cannabinoids, which makes it an excellent candidate to ensure blinding in clinical trials. We investigated, for the first time, the phytochemical profile of this fiber hemp and identified three new compounds: two dihydrophenanthrenes 9, 10 and a quinone 5 closely related to denbinobin 4. Metabololipidomics profiling showed that the three compounds target 5-lipoxygenase and suppress pro-inflammatory LT biosynthesis in activated macrophage subtypes. While each of the compounds might contribute to the resolution of inflammation, their capacity and specificity to elevate the levels of specialized pro-resolving lipid mediators and their precursors significantly differs. Taken together, our results demonstrate the potential of phytochemical and pharmacological characterization of cannabis varieties not only to explain the different biological activities of the chemotypes but also to add to our understanding of the “entourage effect” [5].

## Figures and Tables

**Figure 1 plants-11-02130-f001:**
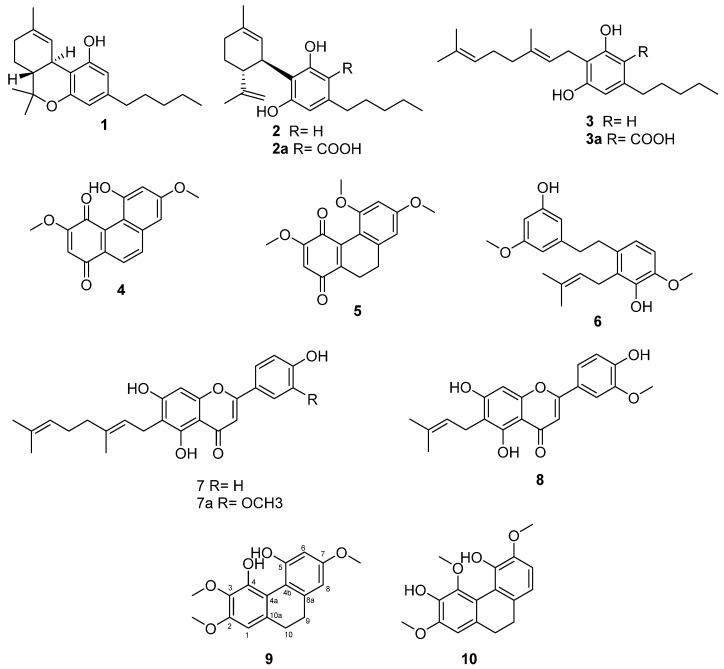
Compounds isolated from *Cannabis sativa* L. Ermo chemotype V.

**Figure 2 plants-11-02130-f002:**
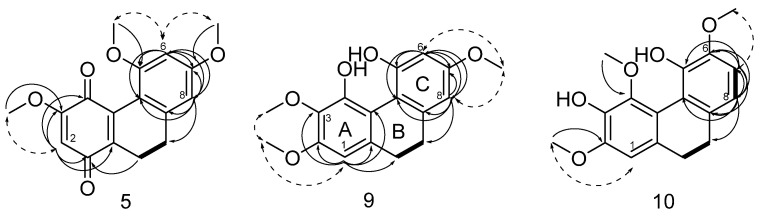
COSY (in bold) and key H→C HMBC (black arrows) and NOESY (dashed arrows) correlations detected for compounds **5**, **9** and **10**.

**Figure 3 plants-11-02130-f003:**
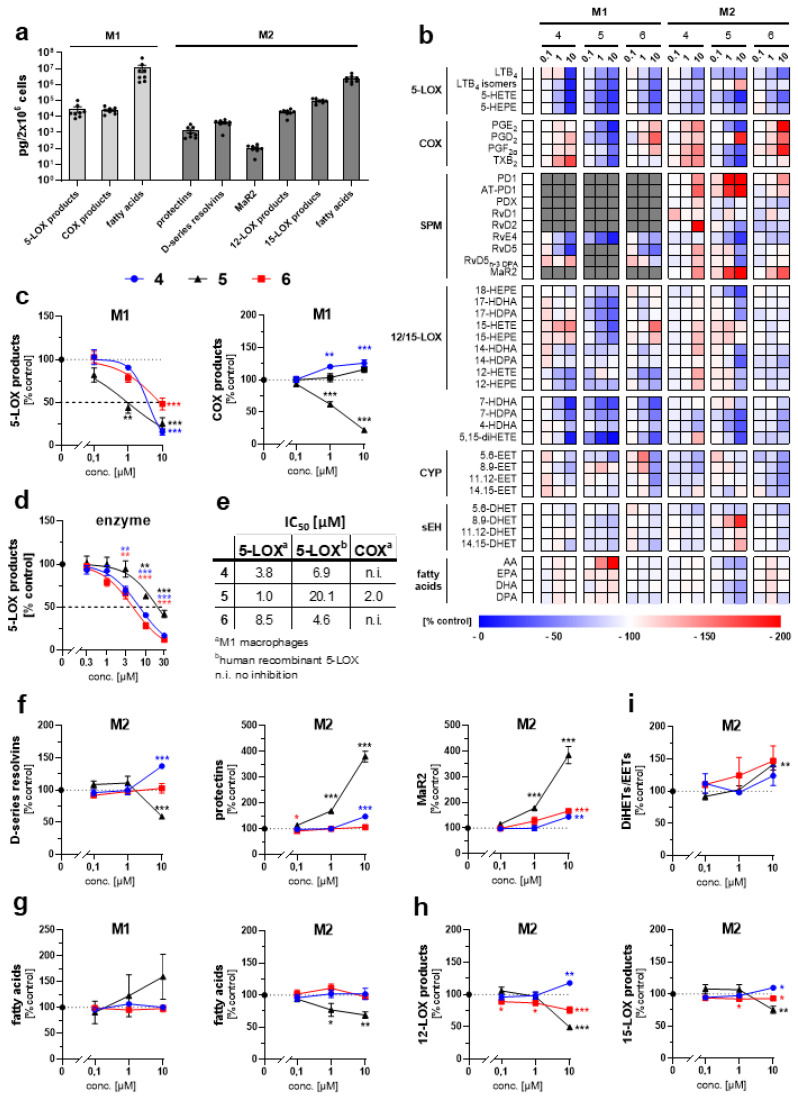
Dihydrophenanthrenoids induce a lipid mediator class switch in activated human M1 and M2 macrophages: (**a**) Absolute amount of lipid mediator subgroups produced by vehicle-treated M1 and M2 macrophages. (**b**) Heatmap showing concentration-dependent changes in the lipid mediator profile. (**c**) 5-LOX and COX products. (**d**) 5-LOX product formation by human recombinant 5-LOX. (**e**) IC_50_ values for the inhibition of 5-LOX or COX product formation in cell-free and cell-based assays. (**f**) SPMs (RvDs, protectins, MaR2). (**g**) Free PUFAs. (**h**) 12- and 15-LOX products. (**i**) Changes in the EETs/DHETs ratio representing the cellular sEH activity. Mean (**b**) or mean ± s.e.m. (**c**,**d**,**f**–**i**) and single data (**a**) from n = 3–4 (**b**–**i**) or n = 8 (**a**) independent experiments. * *p* < 0.05, ** *p* < 0.01, *** *p* < 0.001 vs. vehicle control; repeated measures one-way ANOVA + Dunnett’s post hoc tests of log data.

**Table 1 plants-11-02130-t001:** *Cannabis sativa* L. chemotypes.

Chemotype	Major Cannabinoids	Classification
I	Drug-type plants (narcotic) with high content of the psychotropic Δ^9^-THC	Drug-type plant (narcotic)
II	Medicinal cannabis with Δ^9^-THC/CBD 1:1	Fiber-type
III	Industrial fiber hemp with CBD as predominant and a minimum content of Δ^9^-THC (from 0.2% *w*/*w* to 0.6% *w*/*w*)	Fiber-type
IV	Industrial fiber hemp with CBG as predominant cannabinoid	Fiber-type
V	Industrial fiber hemp with almost no cannabinoids	Fiber-type

**Table 2 plants-11-02130-t002:** ^1^H (400 MHz) NMR data of **5**, **9** and **10** in C_3_D_6_O.

	5	9	10
Position	*δ*_H_, mult (*J* in Hz)	*δ*_H_, mult (*J* in Hz)	*δ*_H_, mult (*J* in Hz)
1		6.62, s	6.63, s
2	5.95, s		
3			
4			
5			
6	6.53, d, 2.40	6.44, d 2.70	
7			6.87, d, 8.2
8	6.53, d, 2.40	6.50, d 2.70	6.82, d, 8.2
9	2.67, t, 7.3	2.64, bs	2.62, bs
10	2.51, t 7.3	2.64, bs	2.62, bs
OMe-2		3.84, s	3.87, s
OMe-3	3.87, s	3.63, s	
OMe-4			3.79, s
OMe-5	3.76, s		
OMe-6			3.86, s
OMe-7	3.85, s	3.81, s	
OH-5		9.09, s	

**Table 3 plants-11-02130-t003:** ^13^C (100 MHz) NMR Data of **5**, **9** and **10** in C_3_D_6_O.

	5	9	10
Position	*δ*_C_, Type	*δ*_C_, Type	*δ*_C_, Type
1	185.2, C	99.1, CH	104.4, CH
2	105.6, CH	151.3, C	152.1, C
3	160.1, C	137.0, C	147.4, C
4	179.2, C	150.3, C	136.6, C
4a	138.0, C	117.3, C	114.7, C
4b	112.0, C	113.3, C	120.7, C
5	158.0, C	155.7, C	141.5, C
6	97.0, CH	102.0, CH	147.6, C
7	162.0, C	160.4, C	109.6, CH
8	105.0, CH	106.4, CH	118.9, CH
8a	138.8, C	127.5, C	132.2, C
9	28.3, CH_2_	31.2, CH_2_	30.3, CH_2_
10	20.1, CH_2_	21.8, CH_2_	31.3, CH_2_
10a	140.1, C	142.7, C	136.7, C
OMe-2		55.3, CH_3_	55.2, CH_3_
OMe-3	55.7, CH_3_	61.2, CH_3_	
OMe-4			59.7, CH_3_
OMe-5	55.2, CH_3_		
OMe-6			55.5, CH_3_
OMe-7	54.9, CH_3_	54.5, CH_3_	

**Table 4 plants-11-02130-t004:** Quantitative MRM transitions. AA, arachidonic acid; AT, aspirin-triggered; EET, epoxyeicosatrienoic acid; EPA, eicosapentaenoic acid; DHA, docosahexaenoic acid; DHET, dihydroxyeicosatrienoic acid; DPA, docosapentaenoic acid; (di)HDHA, (di)hydroxydocosahexaenoic acid, HEPE, hydroxyeicosapentaenoic acid; PD, protectin; TX, thromboxane.

Q1	Q3	ID	DP (V)	EP (V)	CE (eV)	CXP (V)
327.3	116.1	d8-5S-HETE	−80.0	−10.0	−17.0	−10.0
339.3	197.2	d4-LTB_4_	−80.0	−10.0	−22.0	−13.0
355.3	193.2	d4-PGE_2_	−80.0	−10.0	−25.0	−16.0
356.3	115.2	d5-LXA_4_	−80.0	−10.0	−19.0	−14.0
380.3	141.2	d5-RvD_2_	−80.0	−10.0	−23.0	−14.0
311.3	267.1	d8-AA	−100.0	−10.0	−16.0	−18.0
335.2	195.1	LTB_4_ isomers	−80.0	−10.0	−22.0	−13.0
335.2	195.1	LTB_4_	−80.0	−10.0	−22.0	−13.0
319.2	115.1	5-HETE	−80.0	−10.0	−21.0	−12.0
317.2	115.1	5-HEPE	−80.0	−10.0	−18.0	−12.0
351.2	271.0	PGE_2_	−120.0	−10.0	−20.0	−13.0
351.3	233.1	PGD_2_	−80.0	−10.0	−16.0	−15.0
353.3	193.1	PGF_2α_	−80.0	−10.0	−34.0	−11.0
369.3	169.1	TXB_2_	−80.0	−10.0	−22.0	−15.0
375.2	215.1	RvD1	−80.0	−10.0	−26.0	−13.0
375.2	175.1	RvD2	−80.0	−10.0	−30.0	−13.0
333.3	115.1	RvE4	−80.0	−10.0	−22.0	−13.0
359.2	199.1	RvD5	−80.0	−10.0	−21.0	−13.0
361.2	143.0	RvD5_n-3DPA_	−110.0	−10.0	−23.0	−25.0
359.2	250.1	Maresin 1	−80.0	−10.0	−20.0	−16.0
359.2	221.0	Maresin 2	−80.0	−10.0	−20.0	−12.0
359.2	153.1	PD1/PDX/AT-PD1	−80.0	−10.0	−21.0	−9.0
343.2	245.1	17-HDHA	−80.0	−10.0	−17.0	−14.0
345.2	247.1	17-HDPA	−80.0	−10.0	−17.0	−14.0
319.2	219.1	15-HETE	−80.0	−10.0	−19.0	−12.0
317.2	219.1	15-HEPE	−80.0	−10.0	−18.0	−12.0
343.2	205.1	14-HDHA	−80.0	−10.0	−17.0	−14.0
345.2	207.1	14-HDPA	−80.0	−10.0	−17.0	−14.0
319.2	179.1	12-HETE	−80.0	−10.0	−21.0	−12.0
317.2	179.1	12-HEPE	−80.0	−10.0	−19.0	−12.0
317.2	259.1	18-HEPE	−80.0	−10.0	−16.0	−23.0
343.2	141.1	7-HDHA	−80.0	−10.0	−18.0	−15.0
345.2	143.1	7-HDPA	−80.0	−10.0	−18.0	−15.0
343.2	101.1	4-HDHA	−80.0	−10.0	−17.0	−15.0
335.2	201.0	5,15-diHETE	−50.0	−10.0	−30.0	−13.0
319.2	191.1	5.6-EET	−50.0	−5.0	−20.0	−15.0
319.2	167.0	8.9-EET	−40.0	−5.0	−20.0	−10.0
319.2	167.2	11.12-EET	−40.0	−10.0	−20.0	−10.0
319.2	219.2	14.15-EET	−60.0	−10.0	−20.0	−10.0
337.2	145.0	5.6-DHET	−70.0	−5.0	−20.0	−10.0
337.2	127.1	8.9-DHET	−60.0	−5.0	−30.0	−15.0
337.2	167.1	11.12-DHET	−30.0	−5.0	−30.0	−15.0
337.2	207.1	14.15-DHET	−60.0	−5.0	−20.0	−10.0
303.3	259.1	AA	−100.0	−10.0	−16.0	−18.0
301.3	257.1	EPA	−100.0	−10.0	−16.0	−18.0
327.3	283.1	DHA	−100.0	−10.0	−16.0	−18.0
329.3	285.1	DPA	−100.0	−10.0	−16.0	−18.0

## Data Availability

Not applicable.

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
