# Peer review of "Phytochemical Characterization of Cannabis sativa L. Chemotype V Reveals Three New Dihydrophenanthrenoids That Favorably Reprogram Lipid Mediator Biosynthesis in Macrophages"

_plants, 2022, doi:10.3390/plants11162130_

Round 1

Reviewer 1 Report

In their presented work, titled “Phytochemical characterization of Cannabis sativa L. chemotype 2 V reveals three new dihydrophenanthrenoids that favorably reprogram lipid mediator biosynthesis in macrophages”, Salamone et al., investigated the composition of the Cannabis sativa L. Ermo chemotype V and identified new compounds to suppress proinflammatory leukotriene biosynthesis in activated macrophage subtypes.

The manuscript makes an interesting read as scientifically valid methods have been employed to isolate these three new non-cannabinoid compounds from Cannabis sativa, and their biological relevance have demonstrated in proinflammatory regulation. However, the manuscript cannot be published in the current state as it has a significant number of grammatical, typographical and data presentation errors.  It is recommended that the authors seek the professional assistance of an external English editorial office to improve this aspect of the manuscript to the standard required for publication.

The following are some of the errors identified in the review of the manuscript:

(a) Title page: (i) Page 1 line 5-15: “Stefano Salamonea,b,#, Lorenz Waltlc,#, …” The same letters “a, b, c…” should be used in indicating the organizations to which the authors are affiliated.  

(b) Abstract: (i) Page 1 lines 28-29, “dihydro-phenanthenes” should read “dihydro-phenanthrenes”

                        (ii) Page 1 line 19, “Cannabis sativa” should be written in italics i.e. “Cannabis sativa”. And this applies to the rest of the manuscript where this or other scientific names occur. 

(c) Introduction: (i) Page 1 line 39 has the word “spreaded” but the past tense of “spread” is “spread”. 

                            (ii) Page 2 line 46, the word “phenolic” must be changed to “phenolics” or “phenolic compounds” 

                            (iii) Page 2 lines 49-54 reads “I) drug-type plants (narcotic) with a high content of the psychotropic Δ9-tetrahydrocannabinol (Δ9-THC 1); II) medicinal cannabis with Δ9-THC 1/CBD 2 1:1; III) industrial fibre hemp that has CBD 2 as predominant constituent and a minimum content of Δ9-THC 1 (0.2% w/w); chemotype IV fibre-type plants that contain cannabigerol (CBG 3) as the main cannabinoid; and chemotype V, which represents fibre-type plants largely devoid of cannabinoids (Table 1) [4].” 

                            This sentence should be rewritten for clarity, for instance: “The five recognized chemotypes of cannabis include: (I) the drug-type plants (narcotic) with a high content of the psychotropic Δ9-tetrahydrocannabinol (Δ9-THC 1); (II) medicinal cannabis with 1:1 content of Δ9-THC 1:CBD 2; (III) industrial fibre hemp that has CBD 2 as predominant constituent and a minimum content of Δ9-THC 1 (0.2% w/w); (IV) fibre-type plants that contain cannabigerol (CBG 3) as the main cannabinoid; (V) fibre-type plants largely devoid of cannabinoids (Table 1) [4].”

Moreover, the numbers (1, 2, 3 etc.) associated with the compounds should all be either in bold text (or plain) to maintain consistency. Some cannot be written with bold text and others not.

                               (iv) Page 2 lines 64-69 reads “Many hypotheses have been risen about this shortage in producing cannabinoids, which might be the result of the disabling of terpeno-phenolic condensation, a total absence or dysfunctionality of glandular trichomes, but certainly the presence of a cannabinoid knock-out factor that leaves unaffected the biosynthesis of other compound classes inactivating the pathway towards the phenolic cannabinoid precursors [6].” This sentence needs to be re-written in a grammatically correct English to improve clarity.

                              (v) Page 2 lines 89-93, “For this reason, we here investigated for the first time the composition of the Ermo fibre hemp: a variety of hemp with almost no cannabinoids provided by Can-90 vasalus and reported on the isolation, phytochemical characterization, and biological 91 evaluation of phenolic compounds, among them derivatives that have not been described 92 so far.” The meaning of this sentence would be greatly improved if put this way: “For this reason, we here investigated for the first time the composition of the Ermo fibre hemp (a variety of hemp with almost no cannabinoids) provided by Canvasalus and reported on the isolation, phytochemical characterization, and biological evaluation of phenolic compounds, of which three derivatives are new.”

                              (v) Page 2 line 94-95, “immunomodulatory lipid mediator biosynthesis”, should read “immunomodulatory lipid mediator biosynthesis pathways.”

 (d) Results and Discussion: (i) Page 4 line 110, “Morus alba” should be written with italics i.e. “Morus alba

                                               (ii) Page 4 lines 111-113, reads “The Ermo fibre hemp provided three novel natural compounds 5, 9 and 10, which have never been isolated before and belong to the biosynthetic group of dihydrophenantrenoids.” But the word “novel” implies that the compounds “have never been isolated before”, so there is no need for that repetition in the same sentence. Moreover, the word “dihydrophenantrenoids” should read “dihydrophenanthrenoids”.

                                              (iii) Page 4 lines 120-123, “(δC 117.3, 113.3, 127.5 and 142.7) and five oxygenated aromatic quaternary carbons (δC 151.3, 137.0, 150.3, 155.7, 160.4), three aromatic methine carbons (δC 99.1, 102.0, 106.4), two methylene groups (δC 31.2, 21.8) and three methoxyls (δC 55.3, 61.2, 54.5)”. The positions of these carbons should be indicated in the text, same as was done for the protons and methoxy groups before them.

                                              (iv) Page 4 lines 124 to 127, the sentence “In particular, the NOE correlations between the methoxy group at δH 3.84 and H-1 (δH 6.62) and the methoxy groups at δH 3.63 fixed their position at C-2 and C-3, respectively, which was confirmed by HMBC correlations” is not very well explained. It could be rewritten to improve clarity, example “Specifically, NOESY correlations observed between the methoxy group at δH 3.84 and the aromatic proton at H-1 (δH 6.62) together with the NOESY observed between the two methoxy groups at δH 3.84 and δH 3.63, placed the two methoxy groups respectively at positions C-2 and C-3. This was further confirmed by the HMBC observed between H-1 and C-2.”  

(v) Page 4 line 128, “NOESY correlations (H-6/ OMe-7 and H-6/ OMe-7)”, should read “NOESY correlations (H-6/ OMe-7 and H-8/ OMe-7)”

                     (vi) Page 4 lines 136 to 138, “The 13C spectrum (SI) shows the presence again of five oxygenated aromatic quaternary carbons (δC 152.1, 147.4, 136.6, 141.5, 147.6).” The positions of these carbons should be indicated in the text, same as was done for the protons and methoxy groups before them.

                    (v) Page 4 line 139, “and from OMe-6 to H-6”. This HMBC should be rewritten as going from the proton to the carbon i.e. “from H-6 to the carbon of OMe-6)

                    (vi) Page 4 line 139-140, “together with the NOESY correlation OMe-6/H-7 allowed us to identify the structure of ring C.” Should be rewritten as “, together with the NOESY correlation OMe-6/H-7 allowed the identification of the ring C substructure”.

                    (vii) Page 4 line 145, “Its molecular formula was determined to be C17H17O5 with m/z 301.10”. This statement should read, “Its molecular formula was determined by HR-ESIMS to be C17H17O5 with m/z 301.10”. The molecular formulas of compounds 9 and 10 should also be included in the structural description of these compounds.

                 (viii) Page 4 line 156, “H2-10 to C-1”, should read “H-10 to C-1”

                 (ix) Page 5 line 160, “or the novel compound methoxy-dihydrodenbinobin 5”, would read better if written as “or a new 5-methoxy-dihydrodenbinobin derivative 5

                 (x) Page 5 line 161, “Despite the unceasing phytochemical characterisation of cannabis, compounds 5, 9 and 161 10 have no precedent.” Should be rewritten for clarity, e.g. “Despite the fact that a lot of work has been done on the phytochemical characterisation of cannabis, this is the very first time the non-cannabinoid compounds, 5, 9 and 10, have been isolated and characterized from this plant”

                 (xi) Page 5 lines 162-164, “In particular, the dihydrophenanthrenes 9 and 10 retain the same number of methoxy and hydroxyl groups but with different position in the reciprocal scaffold opening the interesting possibility for a structure-relationship study.” This sentence could better be written as “The dihydrophenanthrenes, 9 and 10, are particularly interesting for possible structure-activity relationship study, since both compounds retain the same number of methoxy and hydroxyl groups but with different position in the reciprocal scaffold”.

                   (xii) Page 5 lines 168-170, “Here, we had the opportunity to provide a new chemical entity strictly connected to denbinobin 4, its metoxy-reduced analogue 5.” This phrase should be rewritten in clearer English e.g. “Given the biological relevance of denbinobin 4, the newly isolated methoxy-reduced denbinobin analogue 5 described here is expected to be of similar biological relevance.”

                       (xiii)  Page 5 lines 173-174, “To assess the effect of the here discovered three dihydrophenantrenoids 5, 9 and 10”, would be better rewritten as “To assess the effect of the three new dihydrophenanthrenoids 5, 9 and 10

                        (xiv) Page 5 line 175, “Staphylococcus aureus-”, should be written with Italic text, i.e. “Staphylococcus aureus-”

                        (xv) Page 5 lines 177-180, “Compounds 9 and 10 exhibited anti-inflammatory effects in human M1 macrophages by inhibiting 5-LOX product formation at low micromolar concentrations (Figure 3c), with the quinone 5 being most potent (IC50 = 1.0 μM) and the OH-methylation pattern modulating the 5-LOX product-lowering activity (9: IC50 = 3.8 μM, 10: IC50 = 8.5 μM).” The above sentence is too long and lacks clarity. It may be clearer to understand if split into two different sentences.

                                  (xvi) Page 5 line 188, “whereas the diols 9 and less 10”. What is the meaning of the “less” in this phrase?

                                 (xvii) Page 7, Figure 3f: It´s difficult to tell that the graph of maresin 2 (MaR2) belongs to Figure 3f. This figure should be rearranged to make this clear, or else it may be associated with Figure h by the reader. 

                                (xviii) Page 5 lines 199-201, “It is tempting to speculate that the 3- and/or 7 methoxy-groups, which are shared by 9 and 5 but not present in 10, are required for an effective increase of PDs levels”. This is quite a wide speculation. Are the authors referring to just any 3- and/or 7 methoxy-groups in any class/family of compounds or specifically referring to 3- and/or 7 methoxy-groups in dihydrophenanthrenoids?   

                               (xix) Page 6, line 219, “the here identified dihydrophenanthrenoids”, would sound better as “the newly identified dihydrophenanthrenoids”. 

                              (xx) Page 7 line 252, “Figure 3.” All figure titles and legends should be placed below the figures and not above them (this also applies to Figures 1 and 2).

 (e) Material and Method: (i) Page 8 lines 271-272, “1H−13C”, numbers should be written as superscripts, “1H−13C” 

                                               (ii) Page 8 line 274, “(Thermo Scientific)”. Missing country and city information

                                                (iii) Page 8 line 276, “pression” should be “pressure”

                                               (iv) Page 8 line 277, “Macherey-Nagel”. Missing country and city information

                                               (v) Page 8 line 280-281, “Isolera One 280 with DAD”. Missing company, country, and city information

                                               (vi) Page 8 line 281, “HPLC JASCO Hichrom, 250 × 25 mm, silica UV− vis detector-2075 plus”. Missing company, country, and city information

                                              (vii) Page 8 line 290, “This latter was dissolved”. Do you mean to say “This was later dissolved” or “This latter part (i.e. syrup) was dissolved”?

                                              (viii) Page 8 line 293-294 “This latter has been subsequently purified”, should be “This latter part was subsequently purified”

                                             (ix) Page 8 line 299, “low pression”, should be “low pressure”

                                              (x) Page 8 line 300-301, “to afford three fractions (I, II, III) further purified.”, should be “to afford three fractions (I, II, III), which were further purified”.

                                              (xi) Page 9 line 318, “3.4. Spectroscopic data”. The spectroscopic data of each compound should start on a new line to improve clarity.   

                                              (xii) Page 9 lines 327-329, “Figure 2. COSY (in bold) and key H→C HMBC (black arrows) and NOESY (dashed arrows) correlations detected for compound 5, 9, 10.” This figure should be part of the Results section instead of being in the Material and Method section.

                                                  (xiii) Page 10 line 343, “Table 1. 1H (400 MHz) NMR data of 5, 9 and 10 in C3D6O”. This should be “Table 2” according to the numbering of your tables. And this Table should be part of the Results section instead of the Material and Method section?

                                                    (xiv) Page 11 line 362, “Table 3. 13C (100 MHz) NMR Data of 5, 9 and 10 in C3D6O”, should be part of the Results section instead of being in the Material and Method section.

                                                 (xv) Regarding the statistical analysis, page 13 line 464-465 sates “single data from n = 3-4 independent experiments.” But you also mentioned “n = 8 (a) independent experiments” in the legend of Figure 2 (on page 7 line 259). Why was this not included in the “Statistics” part of the methods?

 (f) Conclusion: (i) Page 15 line 488, “Cannabis sativa”, should be written with italic, i.e. “Cannabis sativa

                           (ii) Page 15 line 491, “dihydrophenantrenes”, should be written as “dihydrophenanthrenes”

 (g) Supplementary Materials: (i) Page 15 lines 508 and 511, the numbers in “C3D6O” should be written as subscripts, i.e. “C3D6O”. 

Author Response

Reviewer #1:

I am really grateful to the reviewer #1 for the great revision made, the deep investigation and the precious corrections and comments to the manuscript.

All the manuscript has been revised to improve and correct English and to avoid mistakes.

(a) Title page: (i) Page 1 line 5-15: “Stefano Salamonea,b,#, Lorenz Waltlc,#, …” The same letters “a, b, c…” should be used in indicating the organizations to which the authors are affiliated.

R: All the affiliations now are indicated with corresponding letters.   

(b) Abstract: (i) Page 1 lines 28-29, “dihydro-phenanthenes” should read “dihydro-phenanthrenes”

R: It has been corrected in all the text.

 (ii) Page 1 line 19, “Cannabis sativa” should be written in italic i.e. “Cannabis sativa”. And this applies to the rest of the manuscript where this or other scientific names occur.

R: It has been corrected in all the text with the itlalics where scientific names occur.

(c) Introduction: (i) Page 1 line 39 has the word “spreaded” but the past tense of “spread” is “spread”.

R: It has been corrected in the text.

(ii) Page 2 line 46, the word “phenolic” must be changed to “phenolics” or “phenolic compounds”

R: It has been corrected with phenolic compounds.

(iii) Page 2 lines 49-54 reads “I) drug-type plants (narcotic) with a high content of the psychotropic Δ9-tetrahydrocannabinol (Δ9-THC 1); II) medicinal cannabis with Δ9-THC 1/CBD 2 1:1; III) industrial fibre hemp that has CBD 2 as predominant constituent and a minimum content of Δ9-THC 1 (0.2% w/w); chemotype IV fibre-type plants that contain cannabigerol (CBG 3) as the main cannabinoid; and chemotype V, which represents fibre-type plants largely devoid of cannabinoids (Table 1) [4].”

This sentence should be rewritten for clarity, for instance: “The five recognized chemotypes of cannabis include: (I) the drug-type plants (narcotic) with a high content of the psychotropic Δ9-tetrahydrocannabinol (Δ9-THC 1); (II) medicinal cannabis with 1:1 content of Δ9-THC 1:CBD 2; (III) industrial fibre hemp that has CBD 2 as predominant constituent and a minimum content of Δ9-THC 1 (0.2% w/w); (IV) fibre-type plants that contain cannabigerol (CBG 3) as the main cannabinoid; (V) fibre-type plants largely devoid of cannabinoids (Table 1) [4].”

Moreover, the numbers (1, 2, 3 etc.) associated with the compounds should all be either in bold text (or plain) to maintain consistency. Some cannot be written with bold text and others not.

R: The sentence has been corrected as the reviewer suggested also with the indication of compounds with bold numbers:

The five recognized chemotypes of cannabis include: (I) the drug-type plants (narcotic) with a high content of the psychotropic Δ9-tetrahydrocannabinol (Δ9-THC 1); (II) medicinal cannabis with 1:1 content of Δ9-THC 1: CBD 2; (III) industrial fibre hemp that has CBD 2 as predominant constituent and a minimum content of Δ9-THC 1 (0.2% w/w); (IV) fibre-type plants that contain cannabigerol (CBG 3) as the main cannabinoid; (V) fibre-type plants largely devoid of cannabinoids (Table 1) [4].

(iv) Page 2 lines 64-69 reads “Many hypotheses have been risen about this shortage in producing cannabinoids, which might be the result of the disabling of terpeno-phenolic condensation, a total absence or dysfunctionality of glandular trichomes, but certainly the presence of a cannabinoid knock-out factor that leaves unaffected the biosynthesis of other compound classes inactivating the pathway towards the phenolic cannabinoid precursors [6].” This sentence needs to be re-written in a grammatically correct English to improve clarity.

R: The sentence has been re-written as follow:

Many hypotheses have been risen about this shortage in producing cannabinoids. For example, the absence of this biosynthesis could be due to the disabling of terpeno-phenolic condensation or a total absence and dysfunctionality of glandular trichomes. Decisive must be the presence of a cannabinoid knock-out factor that leaves unaffected the biosynthesis of other compound classes inactivating the pathway towards the phenolic cannabinoid precursors [6].

 (v) Page 2 lines 89-93, “For this reason, we here investigated for the first time the composition of the Ermo fibre hemp: a variety of hemp with almost no cannabinoids provided by Can-90 vasalus and reported on the isolation, phytochemical characterization, and biological 91 evaluation of phenolic compounds, among them derivatives that have not been described 92 so far.” The meaning of this sentence would be greatly improved if put this way: “For this reason, we here investigated for the first time the composition of the Ermo fibre hemp (a variety of hemp with almost no cannabinoids) provided by Canvasalus and reported on the isolation, phytochemical characterization, and biological evaluation of phenolic compounds, of which three derivatives are new.”

R: The sentence has been re-written as the reviewer suggested.

(v) Page 2 line 94-95, “immunomodulatory lipid mediator biosynthesis”, should read “immunomodulatory lipid mediator biosynthesis pathways.”

R: The terminology has been re-written as the reviewer suggested.

 (d) Results and Discussion: (i) Page 4 line 110, “Morus alba” should be written with italic i.e. “Morus alba”

R: Correction has been made whit italic style.

(ii) Page 4 lines 111-113, reads “The Ermo fibre hemp provided three novel natural compounds 5, 9 and 10, which have never been isolated before and belong to the biosynthetic group of dihydrophenantrenoids.” But the word “novel” implies that the compounds “have never been isolated before”, so there is no need for that repetition in the same sentence. Moreover, the word “dihydrophenantrenoids” should read “dihydrophenanthrenoids”.

R: The repetition novel has been avoid in the text leaving only “have never been isolated before” and the word dihydrophenanthrenoids has been corrected

(iii) Page 4 lines 120-123, “(δC 117.3, 113.3, 127.5 and 142.7) and five oxygenated aromatic quaternary carbons (δC 151.3, 137.0, 150.3, 155.7, 160.4), three aromatic methine carbons (δC 99.1, 102.0, 106.4), two methylene groups (δC 31.2, 21.8) and three methoxyls (δC 55.3, 61.2, 54.5)”. The positions of these carbons should be indicated in the text, same as was done for the protons and methoxy groups before them.

R: The position of every C atoms has been declared as follow:

C 117.3 C-4a, 113.3 C-4b, 127.5 C-8a and 142.7 C-10a) and five oxygenated aromatic quaternary carbons (δC 151.3 C-2, 137.0 C-3, 150.3 C-4, 155.7 C-5, 160.4 C-7), three aromatic methine carbons (δC 99.1 C-1, 102.0 C-6, 106.4 C-8), two methylene groups (δC 31.2 C-9, 21.8 C-10) and three methoxyls (δC 55.3 OMe-2, 61.2 OMe-3, 54.5 OMe-7).

(iv) Page 4 lines 124 to 127, the sentence “In particular, the NOE correlations between the methoxy group at δH 3.84 and H-1 (δH 6.62) and the methoxy groups at δH 3.63 fixed their position at C-2 and C-3, respectively, which was confirmed by HMBC correlations” is not very well explained. It could be rewritten to improve clarity, example “Specifically, NOESY correlations observed between the methoxy group at δH 3.84 and the aromatic proton at H-1 (δH 6.62) together with the NOESY observed between the two methoxy groups at δH 3.84 and δH 3.63, placed the two methoxy groups respectively at positions C-2 and C-3. This was further confirmed by the HMBC observed between H-1 and C-2.” 

R: The sentence has been changed as suggested.

(v) Page 4 line 128, “NOESY correlations (H-6/ OMe-7 and H-6/ OMe-7)”, should read “NOESY correlations (H-6/ OMe-7 and H-8/ OMe-7)”

R: The indication has been changed as suggested.

(vi) Page 4 lines 136 to 138, “The 13C spectrum (SI) shows the presence again of five oxygenated aromatic quaternary carbons (δC 152.1, 147.4, 136.6, 141.5, 147.6).” The positions of these carbons should be indicated in the text, same as was done for the protons and methoxy groups before them.

R: The position of every C atoms of compound 10 has been declared as follow:

The 13C spectrum (SI) shows the presence again of five oxygenated aromatic quaternary carbons (δC 152.1 C-2, 147.4 C-3, 136.6 C-4, 141.5 C-5, 147.6 C-6).

The same has been done for compound 5: The 13C spectrum (SI) showed two carbonylic carbons (δC 185.2 C-1, 179.2 C-4) and confirmed the presence of three sp2 oxygenated carbon atoms (δC 160.1 C-3, 158.0 C-5, 162.0 C-7), while there are still four olefinic disubstituted carbon atoms, two ascribable to the aromatic ring (δC 112.0 C-4b, 138.8 C-8a) and the others to the quinonoid moiety (δC 138.0 C-4a, 140.1 C-10a) and two methines (δC 28.3 C-9, 20.1 C-10).

(v) Page 4 line 139, “and from OMe-6 to H-6”. This HMBC should be rewritten as going from the proton to the carbon i.e. “from H-6 to the carbon of OMe-6)

R: The indication has been re-written as suggested.

(vi) Page 4 line 139-140, “together with the NOESY correlation OMe-6/H-7 allowed us to identify the structure of ring C.” Should be rewritten as “, together with the NOESY correlation OMe-6/H-7 allowed the identification of the ring C substructure”.

R: The sentence has been re-written as the reviewer suggested.

(vii) Page 4 line 145, “Its molecular formula was determined to be C17H17O5 with m/z 301.10”. This statement should read, “Its molecular formula was determined by HR-ESIMS to be C17H17O5 with m/z 301.10”. The molecular formulas of compounds 9 and 10 should also be included in the structural description of these compounds.

R: The indication has been provided for compound 9 and 10 in the text (C17H19O5 with m/z 303.12 [M + H]+ )

(viii) Page 4 line 156, “H2-10 to C-1”, should read “H-10 to C-1”

R: Indication has been changed.

(ix) Page 5 line 160, “or the novel compound methoxy-dihydrodenbinobin 5”, would read better if written as “or a new 5-methoxy-dihydrodenbinobin derivative 5”

R: The indication has been changed as suggested

(x) Page 5 line 161, “Despite the unceasing phytochemical characterisation of cannabis, compounds 5, 9 and 161 10 have no precedent.” Should be rewritten for clarity, e.g. “Despite the fact that a lot of work has been done on the phytochemical characterisation of cannabis, this is the very first time the non-cannabinoid compounds, 5, 9 and 10, have been isolated and characterized from this plant”

R: The sentence has been changed as the reviewer indicated

(xi) Page 5 lines 162-164, “In particular, the dihydrophenanthrenes 9 and 10 retain the same number of methoxy and hydroxyl groups but with different position in the reciprocal scaffold opening the interesting possibility for a structure-relationship study.” This sentence could better be written as “The dihydrophenanthrenes, 9 and 10, are particularly interesting for possible structure-activity relationship study, since both compounds retain the same number of methoxy and hydroxyl groups but with different position in the reciprocal scaffold”.

R: The sentence has been changed as the reviewer indicated

 (xii) Page 5 lines 168-170, “Here, we had the opportunity to provide a new chemical entity strictly connected to denbinobin 4, its metoxy-reduced analogue 5.” This phrase should be rewritten in clearer English e.g. “Given the biological relevance of denbinobin 4, the newly isolated methoxy-reduced denbinobin analogue 5 described here is expected to be of similar biological relevance.”

R: The sentence has been changed as the reviewer indicated

(xiii)  Page 5 lines 173-174, “To assess the effect of the here discovered three dihydrophenantrenoids 5, 9 and 10”, would be better rewritten as “To assess the effect of the three new dihydrophenanthrenoids 5, 9 and 10”

R: The sentence has been changed as the reviewer indicated

(xiv) Page 5 line 175, “Staphylococcus aureus-”, should be written with Italic text, i.e. “Staphylococcus aureus-”

R: It has been corrected in italic style

 (xv) Page 5 lines 177-180, “Compounds 9 and 10 exhibited anti-inflammatory effects in human M1 macrophages by inhibiting 5-LOX product formation at low micromolar concentrations (Figure 3c), with the quinone 5 being most potent (IC50 = 1.0 μM) and the OH-methylation pattern modulating the 5-LOX product-lowering activity (9: IC50 = 3.8 μM, 10: IC50 = 8.5 μM).” The above sentence is too long and lacks clarity. It may be clearer to understand if split into two different sentences.

R: The sentence has been divided as:

Compounds 9 and 10 exhibited anti-inflammatory effects in human M1 macrophages by inhibiting 5-LOX product formation at low micromolar concentrations. (Figure 3c). The quinone 5 showed major activity (IC50 = 1.0 µM) with the OH-methylation pattern modu-lating the 5-LOX product-lowering activity (9: IC50 = 3.8 µM, 10: IC50 = 8.5 µM).

(xvi) Page 5 line 188, “whereas the diols 9 and less 10”. What is the meaning of the “less” in this phrase?

R: It was a typing error, the word less has been deleted in the sentence.

(xvii) Page 7, Figure 3f: It´s difficult to tell that the graph of maresin 2 (MaR2) belongs to Figure 3f. This figure should be rearranged to make this clear, or else it may be associated with Figure h by the reader.

R: In Fig. 3, the position of letters indicating figures has been rearranged in order to have a better identification.

(xviii) Page 5 lines 199-201, “It is tempting to speculate that the 3- and/or 7 methoxy-groups, which are shared by 9 and 5 but not present in 10, are required for an effective increase of PDs levels”. This is quite a wide speculation. Are the authors referring to just any 3- and/or 7 methoxy-groups in any class/family of compounds or specifically referring to 3- and/or 7 methoxy-groups in dihydrophenanthrenoids?

R: The authors are referring only to these specific compounds 9 and 10.

It is tempting to speculate that the 3- and/or 7-methoxy-groups, which are shared by 9 and 5 but not present in 10 in these isolated compounds, are required for an effective increase of PDs levels.

(xix) Page 6, line 219, “the here identified dihydrophenanthrenoids”, would sound better as “the newly identified dihydrophenanthrenoids”.

R: Changed as suggested.

 (xx) Page 7 line 252, “Figure 3.” All figure titles and legends should be placed below the figures and not above them (this also applies to Figures 1 and 2).

R: Legends of all figures have been placed below

 (e) Material and Method: (i) Page 8 lines 271-272, “1H−13C”, numbers should be written as superscripts, “1H−13C”

R: Corrected in the text

(ii) Page 8 line 274, “(Thermo Scientific)”. Missing country and city information

R: Added in the text

 (iii) Page 8 line 276, “pression” should be “pressure”

R: Corrected with pressure

 (iv) Page 8 line 277, “Macherey-Nagel”. Missing country and city information

R: Added in the text

 (v) Page 8 line 280-281, “Isolera One 280 with DAD”. Missing company, country, and city information

R: Added in the text

 (vi) Page 8 line 281, “HPLC JASCO Hichrom, 250 × 25 mm, silica UV− vis detector-2075 plus”. Missing company, country, and city information

R: Added in the text

 (vii) Page 8 line 290, “This latter was dissolved”. Do you mean to say “This was later dissolved” or “This latter part (i.e. syrup) was dissolved”?

R: Completed wit “this latter part”

(viii) Page 8 line 293-294 “This latter has been subsequently purified”, should be “This latter part was subsequently purified”

R: Corrected as suggested.

(ix) Page 8 line 299, “low pression”, should be “low pressure”

R: Corrected in the text

(x) Page 8 line 300-301, “to afford three fractions (I, II, III) further purified.”, should be “to afford three fractions (I, II, III), which were further purified”.

R: Corrected in the text

 (xi) Page 9 line 318, “3.4. Spectroscopic data”. The spectroscopic data of each compound should start on a new line to improve clarity.  

R: Each compound has a new line.

(xii) Page 9 lines 327-329, “Figure 2. COSY (in bold) and key H→C HMBC (black arrows) and NOESY (dashed arrows) correlations detected for compound 5, 9, 10.” This figure should be part of the Results section instead of being in the Material and Method section.

R: The position of figure 2 has been moved to the results section

 (xiii) Page 10 line 343, “Table 1. 1H (400 MHz) NMR data of 5, 9 and 10 in C3D6O”. This should be “Table 2” according to the numbering of your tables. And this Table should be part of the Results section instead of the Material and Method section?

R: The position of table 2 has been moved to the results section

 (xiv) Page 11 line 362, “Table 3. 13C (100 MHz) NMR Data of 5, 9 and 10 in C3D6O”, should be part of the Results section instead of being in the Material and Method section.

R: The position of table 3 has been moved to the results section

 (xv) Regarding the statistical analysis, page 13 line 464-465 sates “single data from n = 3-4 independent experiments.” But you also mentioned “n = 8 (a) independent experiments” in the legend of Figure 2 (on page 7 line 259). Why was this not included in the “Statistics” part of the methods?

R: It has been corrected with: single data from 4-8 independent experiments and added to the “Statistics”

 (f) Conclusion: (i) Page 15 line 488, “Cannabis sativa”, should be written with italic, i.e. “Cannabis sativa”

R: It has been corrected in italic

(ii) Page 15 line 491, “dihydrophenantrenes”, should be written as “dihydrophenanthrenes”

R: It has been corrected in the text

 (g) Supplementary Materials: (i) Page 15 lines 508 and 511, the numbers in “C3D6O” should be written as subscripts, i.e. “C3D6O”.

R: It has been corrected in the text

Reviewer 2 Report

In this manuscript, the authors obtain the phytochemical profile of Cannabis sativa L. chemotype V. The article is well organized and of great interest since it is the first study on its chemical composition and three new compounds with biological activities of interest have been identified. 

However, I have some comments to suggest related to the improvement of the formal aspects of the paper.

- The affiliations of the authors must be corrected, letters and numbers appear (?) Also in keywords, the numbers must be eliminated

- The legends of the figures are not in the correct position.

- I think that in general the text would be easier to read, beginning with the introduction, through the use of paragraphs.

- In my opinion, figure 3 shows too much information. Perhaps it would be appropriate to divide it into several, which would also allow a better visualization of the data. I would also consider the fact that this information is. very detailed in the text

- The bibliography shows a different font than the rest of the text.

Author Response

I am really grateful to the reviewers and to you for the time dedicated to our manuscript.

All issues raised by the reviewers have been taken into considerations, and discussed with a point-by point responses to comments, corrections and suggestions.

Reviewer #2:

I am really grateful to the reviewer #2 for the suggestions made.

The affiliations of the authors must be corrected, letters and numbers appear (?) Also in keywords, the numbers must be eliminated

R: The affiliations have been corrected with letters as authors

- The legends of the figures are not in the correct position.

R: Legends of all figures have been positioned below every figures.

- I think that in general the text would be easier to read, beginning with the introduction, through the use of paragraphs.

R: The manuscript has been divided in paragraph as the reviewer kindly suggested. Moreover, figures have been distributed to divide spectroscopic results from biological results.

- In my opinion, figure 3 shows too much information. Perhaps it would be appropriate to divide it into several, which would also allow a better visualization of the data. I would also consider the fact that this information is. very detailed in the text

R: Figure 3 has been improved in dimension to allow a better visualization of all data

- The bibliography shows a different font than the rest of the text.

R: The biography has been changed in the article style required.

Reviewer 3 Report

The manuscript entitled : Phytochemical characterization of Cannabis sativa L. chemotype V reveals three new dihydrophenanthrenoids that favorably re- program lipid mediator biosynthesis in macrophages investigated the phytochemical composition of Cannabis sativa L. chemotype V and discovered compounds that might promote inflammation resolution by promoting a lipid mediator class switch.

The manuscript is well designed, well written and adequately discussed according to the literature data. Appropriate methods were used in the study.

The manuscript can be accepted in the present form.

Author Response

I am really grateful to the reviewer #3 for the positive opinion on the paper